# RETHINKING EMBEDDING COUPLING
# IN PRE-TRAINED LANGUAGE MODELS

**Hyung Won Chung**[*†]
Google Research
hwchung@google.com

**Thibault Févry**[*†]
thibaultfevry@gmail.com

**Henry Tsai**
Google Research
henrytsai@google.com

**Melvin Johnson**
Google Research
melvinp@google.com

**Sebastian Ruder**
DeepMind
ruder@google.com

## ABSTRACT

We re-evaluate the standard practice of sharing weights between input and output embeddings in state-of-the-art pre-trained language models. We show that decoupled embeddings provide increased modeling flexibility, allowing us to significantly improve the efficiency of parameter allocation in the input embedding of multilingual models. By reallocating the input embedding parameters in the Transformer layers, we achieve dramatically better performance on standard natural language understanding tasks with the same number of parameters during fine-tuning. We also show that allocating additional capacity to the output embedding provides benefits to the model that persist through the fine-tuning stage even though the output embedding is discarded after pre-training. Our analysis shows that larger output embeddings prevent the model's last layers from overspecializing to the pre-training task and encourage Transformer representations to be more general and more transferable to other tasks and languages. Harnessing these findings, we are able to train models that achieve strong performance on the XTREME benchmark without increasing the number of parameters at the fine-tuning stage.

## 1 INTRODUCTION

The performance of models in natural language processing (NLP) has dramatically improved in recent years, mainly driven by advances in transfer learning from large amounts of unlabeled data (Howard & Ruder, 2018; Devlin et al., 2019). The most successful paradigm consists of pre-training a large Transformer (Vaswani et al., 2017) model with a self-supervised loss and fine-tuning it on data of a downstream task (Ruder et al., 2019). Despite its empirical success, inefficiencies have been observed related to the training duration (Liu et al., 2019b), pre-training objective (Clark et al., 2020b), and training data (Conneau et al., 2020a), among others. In this paper, we reconsider a modeling assumption that may have a similarly pervasive practical impact: the coupling of input and output embeddings[1] in state-of-the-art pre-trained language models.

State-of-the-art pre-trained language models (Devlin et al., 2019; Liu et al., 2019b) and their multilingual counterparts (Devlin et al., 2019; Conneau et al., 2020a) have inherited the practice of embedding coupling from their language model predecessors (Press & Wolf, 2017; Inan et al., 2017). However, in contrast to their language model counterparts, embedding coupling in encoder-only pre-trained models such as Devlin et al. (2019) is only useful during pre-training since output embeddings are generally discarded after fine-tuning.[2] In addition, given the willingness of researchers to exchange additional compute during pre-training for improved downstream performance (Raffel

---

[*]equal contribution

[†]Work done as a member of the Google AI Residency Program.

[1]Output embedding is sometimes referred to as "output weights", i.e., the weight matrix in the output projection in a language model.

[2]We focus on encoder-only models, and do not consider encoder-decoder models like T5 (Raffel et al., 2020) where none of the embedding matrices are discarded after pre-training. Output embeddings may also be

Table 1: Overview of the number of parameters in (coupled) embedding matrices of state-of-the-art multilingual (top) and monolingual (bottom) models with regard to overall parameter budget. $|V|$: vocabulary size. $N$, $N_{\text{emb}}$: number of parameters in total and in the embedding matrix respectively.

| Model | Languages | $|V|$ | $N$ | $N_{\text{emb}}$ | %Emb. |
|---|---|---|---|---|---|
| mBERT (Devlin et al., 2019) | 104 | 120k | 178M | 92M | 52% |
| XLM-R$_{\text{Base}}$ (Conneau et al., 2020a) | 100 | 250k | 270M | 192M | 71% |
| XLM-R$_{\text{Large}}$ (Conneau et al., 2020a) | 100 | 250k | 550M | 256M | 47% |
| BERT$_{\text{Base}}$ (Devlin et al., 2019) | 1 | 30k | 110M | 23M | 21% |
| BERT$_{\text{Large}}$ (Devlin et al., 2019) | 1 | 30k | 335M | 31M | 9% |

et al., 2020; Brown et al., 2020) and the fact that pre-trained models are often used for inference millions of times (Wolf et al., 2019), pre-training-specific parameter savings are less important overall.

On the other hand, tying input and output embeddings constrains the model to use the same dimensionality for both embeddings. This restriction limits the researcher's flexibility in parameterizing the model and can lead to allocating too much capacity to the input embeddings, which may be wasteful. This is a problem particularly for multilingual models, which require large vocabularies with high-dimensional embeddings that make up between 47–71% of the entire parameter budget (Table 1), suggesting an inefficient parameter allocation.

In this paper, we systematically study the impact of embedding coupling on state-of-the-art pretrained language models, focusing on multilingual models. First, we observe that while naïvely decoupling the input and output embedding *parameters* does not consistently improve downstream evaluation metrics, decoupling their *shapes* comes with a host of benefits. In particular, it allows us to independently modify the input and output embedding dimensions. We show that the input embedding dimension can be safely reduced without affecting downstream performance. Since the output embedding is discarded after pre-training, we can increase its dimension, which improves fine-tuning accuracy and outperforms other capacity expansion strategies. By reinvesting saved parameters to the width and depth of the Transformer layers, we furthermore achieve significantly improved performance over a strong mBERT (Devlin et al., 2019) baseline on multilingual tasks from the XTREME benchmark (Hu et al., 2020). Finally, we combine our techniques in a Rebalanced mBERT (RemBERT) model that outperforms XLM-R (Conneau et al., 2020a), the state-of-the-art cross-lingual model while having been pre-trained on $3.5\times$ fewer tokens and 10 more languages.

We thoroughly investigate reasons for the benefits of embedding decoupling. We observe that an increased output embedding size enables a model to improve on the pre-training task, which correlates with downstream performance. We also find that it leads to Transformers that are more transferable across tasks and languages—particularly for the upper-most layers. Overall, larger output embeddings prevent the model's last layers from over-specializing to the pre-training task (Zhang et al., 2020; Tamkin et al., 2020), which enables training of more general Transformer models.

## 2 RELATED WORK

**Embedding coupling**    Sharing input and output embeddings in neural language models was proposed to improve perplexity and motivated based on embedding similarity (Press & Wolf, 2017) as well as by theoretically showing that the output probability space can be constrained to a subspace governed by the embedding matrix for a restricted case (Inan et al., 2017). Embedding coupling is also common in neural machine translation models where it reduces model complexity (Firat et al., 2016) and saves memory (Johnson et al., 2017), in recent state-of-the-art language models (Melis et al., 2020), as well as all pre-trained models we are aware of (Devlin et al., 2019; Liu et al., 2019b).

**Transferability of representations**    Representations of large pre-trained models in computer vision and NLP have been observed to transition from general to task-specific from the first to the

_______________________

useful for domain-adaptive pre-training (Howard & Ruder, 2018; Gururangan et al., 2020), probing (Elazar & Goldberg, 2019), and tasks that can be cast in the pre-training objective (Amrami & Goldberg, 2019).

last layer (Yosinski et al., 2014; Howard & Ruder, 2018; Liu et al., 2019a). In Transformer models, the last few layers have been shown to become specialized to the MLM task and—as a result—less transferable (Zhang et al., 2020; Tamkin et al., 2020).

**Multilingual models**   Recent multilingual models are pre-trained on data covering around 100 languages using a subword vocabulary shared across all languages (Devlin et al., 2019; Pires et al., 2019; Conneau et al., 2020a). In order to achieve reasonable performance for most languages, these models need to allocate sufficient capacity for each language, known as the curse of multilinguality (Conneau et al., 2020a; Pfeiffer et al., 2020). As a result, such multilingual models have large vocabularies with large embedding sizes to ensure that tokens in all languages are adequately represented.

**Efficient models**   Most work on more efficient pre-trained models focuses on pruning or distillation (Hinton et al., 2015). Pruning approaches remove parts of the model, typically attention heads (Michel et al., 2019; Voita et al., 2019) while distillation approaches distill a large pre-trained model into a smaller one (Sun et al., 2020). Distillation can be seen as an alternative form of allocating pre-training capacity via a large teacher model. However, distilling a pre-trained model is expensive (Sanh et al., 2019) and requires overcoming architecture differences and balancing training data and loss terms (Mukherjee & Awadallah, 2020). Our proposed methods are simpler and complementary to distillation as they can improve the pre-training of compact student models (Turc et al., 2019).

## 3   EXPERIMENTAL METHODOLOGY

Efficiency of models has been measured along different dimensions, from the number of floating point operations (Schwartz et al., 2019) to their runtime (Zhou et al., 2020). We follow previous work (Sun et al., 2020) and compare models in terms of their number of parameters during fine-tuning (see Appendix A.1 for further justification of this setting). For completeness, we generally report the number of pre-training (PT) and fine-tuning (FT) parameters.

**Baseline**   Our baseline has the same architecture as multilingual BERT (mBERT; Devlin et al., 2019). It consists of 12 Transformer layers with a hidden size $H$ of 768. Input and output embeddings are coupled and have the same dimensionality $E$ as the hidden size, i.e. $E_{\text{out}} = E_{\text{in}} = H$. The total number of parameters during pre-training and fine-tuning is 177M (see Appendix A.2 for further details). We train variants of this model that differ in certain hyper-parameters but otherwise are trained under the same conditions to ensure a fair comparison.

**Tasks**   For our experiments, we employ tasks from the XTREME benchmark (Hu et al., 2020) that require fine-tuning, including the XNLI (Conneau et al., 2018), NER (Pan et al., 2017), PAWS-X (Yang et al., 2019), XQuAD (Artetxe et al., 2020), MLQA (Lewis et al., 2020), and TyDiQA-GoldP (Clark et al., 2020a) datasets. We provide details for them in Appendix A.4. We average results across three fine-tuning runs and evaluate on the dev sets unless otherwise stated.

## 4   EMBEDDING DECOUPLING REVISITED

**Naïve decoupling**   Embeddings make up a large fraction of the parameter budget in state-of-the-art multilingual models (see Table 1). We now study the effect of embedding decoupling on such models. In Table 2, we show the impact of decoupling the input and output embeddings in our baseline model (§3) with coupled embeddings. Naïvely decoupling the output embedding matrix slightly improves the performance as evidenced by a 0.4 increase on average. However, the gain is not uniformly observed in all tasks. Overall, these results suggest that decoupling the embedding matrices naïvely while keeping the dimensionality fixed does not greatly affect the performance of the model. What is more important, however, is that decoupling the input and output embeddings decouples the *shapes*, endowing significant modeling flexibility, which we investigate in the following.

**Input vs output embeddings**   Decoupling input and output embeddings allows us to flexibly change the dimensionality of both matrices and to determine which one is more important for good transfer performance of the model. To this end, we compare the performance of a model with

Table 2: Effect of decoupling the input and output embedding matrices on performance on multiple tasks in XTREME. PT: Pre-training. FT: Fine-tuning. The decoupled model has input and output embeddings with the same size ($E = 768$) as the embedding of the coupled model. The Transformer parts of the models are the same (i.e., 12 layers with $H = 768$).

| | # PT params | # FT params | XNLI Acc | NER F1 | PAWS-X Acc | XQuAD EM/F1 | MLQA EM/F1 | TyDi-GoldP EM/F1 | Avg |
|---|---|---|---|---|---|---|---|---|---|
| Coupled | 177M | 177M | 70.7 | **69.2** | **85.3** | 46.2/63.2 | **37.3/53.1** | 40.7/56.7 | 62.3 |
| Decoupled | 269M | 177M | **71.3** | 68.9 | 85.0 | **46.9/63.8** | **37.3/53.1** | **42.8/58.1** | **62.7** |

Table 3: Performance of models with a large input and small output embedding size and vice versa. Both models have 12 Transformer layers with $H = 768$.

| | # PT params | # FT params | XNLI Acc | NER F1 | PAWS-X Acc | XQuAD EM/F1 | MLQA EM/F1 | TyDi-GoldP EM/F1 | Avg |
|---|---|---|---|---|---|---|---|---|---|
| $E_{\text{in}} = 768$, $E_{\text{out}} = 128$ | 192M | 177M | 70.0 | **68.3** | 84.3 | 42.0/**60.8** | **34.7/50.9** | 35.2/**52.2** | 60.1 |
| $E_{\text{in}} = 128$, $E_{\text{out}} = 768$ | 192M | 100M | **70.4** | 67.6 | **84.9** | **43.9**/60.0 | 34.6/49.5 | **37.8**/51.0 | **60.2** |

$E_{\text{in}} = 768$, $E_{\text{out}} = 128$ to that of a model with $E_{\text{in}} = 128$, $E_{\text{out}} = 768$[3] (the remaining hyper-parameters are the same as the baseline in §3). During fine-tuning, the latter model has 43% fewer parameters. We show the results in Table 3. Surprisingly, the model pre-trained with a larger output embedding size is competitive with the comparison method on average despite having 77M fewer parameters during fine-tuning.[4]

Reducing the input embedding dimension saves a significant number of parameters at a noticeably smaller cost to accuracy than reducing the output embedding size. In light of this, the parameter allocation of multilingual models (see Table 1) seems particularly inefficient. For a multilingual model with coupled embeddings, reducing the input embedding dimension to save parameters as proposed by Lan et al. (2020) is very detrimental to performance (see Appendix A.5 for details).

The results in this section indicate that the output embedding plays an important role in the transferability of pre-trained representations. For multilingual models in particular, a small input embedding dimension frees up a significant number of parameters at a small cost to performance. In the next section, we study how to improve the performance of a model by resizing embeddings and layers.

## 5 EMBEDDING AND LAYER RESIZING FOR MORE EFFICIENT FINE-TUNING

**Increasing the output embedding size** In §4, we observed that reducing $E_{\text{out}}$ hurts performance on the fine-tuning tasks, suggesting $E_{\text{out}}$ is important for transferability. Motivated by this result, we study the opposite scenario, i.e., whether increasing $E_{\text{out}}$ beyond $H$ improves the performance. We experiment with an output embedding size $E_{\text{out}}$ in the range $\{128, 768, 3072\}$ while keeping the input embedding size $E_{\text{in}} = 128$ and all other parts of the model the same as described in §3 ($H = 768$, 12 layers, etc).

We show the results in Table 4. In all of the tasks we consider, increasing $E_{\text{out}}$ monotonically improves the performance. The improvement is particularly impressive for the more complex question answering datasets. It is important to note that during fine-tuning, all three models have *the exact same sizes* for $E_{\text{in}}$ and $H$. The only difference among them is the output embedding, which is discarded after pre-training. These results show that the effect of additional capacity during pre-training persists through the fine-tuning stage even if the added capacity is discarded after pre-training. We perform an extensive analysis on this behavior in §6. We show results with an English BERT$_{\text{Base}}$ model in Appendix A.6, which show the same trend.

---

[3]We linearly project the embeddings from $E_{\text{in}}$ to $H$ and from $H$ to $E_{\text{out}}$.

[4]We observe the same trend if we control for the number of *trainable parameters* during fine-tuning by freezing the input embedding parameters.

Table 4: Effect of an increased output embedding size $E_{\text{out}}$ on tasks in XTREME. All three models have $E_{\text{in}} = 128$ and 12 Transformer layers with $H = 768$.

| | # PT params | # FT params | XNLI Acc | NER F1 | PAWS-X Acc | XQuAD EM/F1 | MLQA EM/F1 | TyDi-GoldP EM/F1 | Avg |
|---|---|---|---|---|---|---|---|---|---|
| $E_{\text{out}} = 128$ | 115M | 100M | 68.1 | 65.2 | 83.3 | 38.6/54.8 | 30.9/45.2 | 32.2/44.2 | 56.6 |
| $E_{\text{out}} = 768$ | 192M | 100M | 70.4 | 67.6 | 84.9 | 43.9/60.0 | 34.6/49.5 | 37.8/51.0 | 60.2 |
| $E_{\text{out}} = 3072$ | 469M | 100M | **71.1** | **68.1** | **85.1** | **45.3/63.3** | **37.2/53.1** | **39.4/54.7** | **61.8** |

Table 5: Effect of additional capacity via more Transformer layers during pre-training. Both models have $E_{\text{in}} = 128$. The $E_{\text{out}} = 768$ model has a larger output embedding size $E_{\text{out}}$ and 12 Transformer layers. In contrast, the model with 11 additional Transformer layers has $E_{\text{out}} = 128$. Those additional layers are dropped after pre-training, leaving 12 layers for fair comparison during fine-tuning.

| | # PT params | # FT params | XNLI Acc | NER F1 | PAWS-X Acc | XQuAD EM/F1 | MLQA EM/F1 | TyDi-GoldP EM/F1 | Avg |
|---|---|---|---|---|---|---|---|---|---|
| $E_{\text{out}} = 768$ | 192M | 100M | 70.4 | **67.6** | 84.9 | **43.9/60.0** | **34.6/49.5** | **37.8/51.0** | **60.2** |
| 11 add. layers | 193M | 100M | **71.2** | 67.3 | **85.0** | 38.8/55.5 | 31.4/46.6 | 31.3/45.5 | 58.0 |

**Adding capacity via layers** We investigate alternative ways of adding capacity during pre-training such as increasing the number of layers and discarding them after pre-training. For a fair comparison with the $E_{\text{out}} = 768$ model, we add 11 additional layers (total of 23) and drop the 11 upper layers after pre-training. This setting ensures that both models have the same pre-training and fine-tuning parameters. We show the results in Table 5. The model with additional layers performs poorly on the question answering tasks, likely because the top layers contain useful semantic information (Tenney et al., 2019). In addition to higher performance, increasing $E_{\text{out}}$ relies only a more expensive dense matrix multiplication, which is highly optimized on typical accelerators and can be scaled up more easily with model parallelism (Shazeer et al., 2018) because of small additional communication cost. We thus focus on increasing $E_{\text{out}}$ to expand pre-training capacity and leave an exploration of alternative strategies to future work.

**Reinvesting input embedding parameters** Reducing $E_{\text{in}}$ from 768 to 128 reduces the number of parameters from 177M to 100M. We redistribute these 77M parameters for the model with $E_{\text{out}} = 768$ to add capacity where it might be more useful by increasing the width or depth of the model. Specifically, we 1) increase the hidden dimension $H$ of the Transformer layers from 768 to 1024[5] and 2) increase the number of Transformer layers ($L$) from 12 to 23 at the same $H$ to obtain models with similar number of parameters during fine-tuning.

Table 6 shows the results for these two strategies. Reinvesting the input embedding parameters in both $H$ and $L$ improves performance on all tasks while increasing the number of Transformer layers $L$ results in the best performance, with an average improvement of 3.9 over the baseline model with coupled embeddings and the same number of fine-tuning parameters overall.

**A rebalanced mBERT** We finally combine and scale up our techniques to design a rebalanced mBERT model that outperforms the current state-of-the-art unsupervised model, XLM-R (Conneau et al., 2020a). As the performance of Transformer-based models strongly depends on their number of parameters (Raffel et al., 2020), we propose a Rebalanced mBERT (RemBERT) model that matches XLM-R's number of fine-tuning parameters (559M) while using a reduced embedding size, resized layers, and more effective capacity during pre-training. The model has a vocabulary size of 250k, $E_{\text{in}} = 256$, $E_{\text{out}} = 1536$, and 32 layers with 1152 dimensions and 18 attention heads per layer and was trained on data covering 110 languages. We provide further details in Appendix A.7.

We compare RemBERT to XLM-R and the best-performing models on the XTREME leaderboard in Table 7 (see Appendix A.8 for the per-task results).[6] The models in the first three rows use

---

[5]We choose 1024 dimensions to optimize efficient use of our accelerators.

[6]We do not consider retrieval tasks as they require intermediate task data (Phang et al., 2020).

Table 6: Effect of reinvesting the input embedding parameters to increase the hidden dimension $H$ and number of Transformer layers $L$ on XTREME tasks. $E_{in} = 128, E_{out} = 768, H = 768$ for all models except for the baseline, which has coupled embeddings and $E_{in} = E_{out} = 768$.

| | # PT params | # FT params | XNLI Acc | NER F1 | PAWS-X Acc | XQuAD EM/F1 | MLQA EM/F1 | TyDi-GoldP EM/F1 | Avg |
|---|---|---|---|---|---|---|---|---|---|
| Baseline | 177M | 177M | 70.7 | 69.2 | 85.3 | 46.2/63.2 | 37.3/53.1 | 40.7/56.7 | 62.3 |
| $E_{in} = 128, E_{out} = 768$ | 192M | 100M | 70.4 | 67.6 | 84.9 | 43.9/60.0 | 34.6/49.5 | 37.8/51.0 | 60.2 |
| Reinvested in $H$ | 260M | 168M | 72.8 | 69.2 | 85.6 | 50.2/67.2 | 40.7/56.4 | 44.8/60.0 | 64.5 |
| Reinvested in $L$ | 270M | 178M | **73.6** | **71.0** | **86.7** | **51.7/68.8** | **42.4/58.2** | **48.2/62.9** | **66.2** |

Table 7: Comparison of our model to other models on the XTREME leaderboard. Details about VECO are due to communication with the authors.

| | # PT params | # FT params | Langs | Add. task data | Trans-lation data | Sentence-pair Classification Acc | Structured Prediction F1 | Question Answering EM/F1 | Avg |
|---|---|---|---|---|---|---|---|---|---|
| *Models fine-tuned on translations or additional task data* | | | | | | | | | |
| STiLTs (Phang et al., 2020) | 559M | 559M | 100 | ✓ | | 83.9 | 69.4 | 67.2 | 73.5 |
| FILTER (Fang et al., 2020) | 559M | 559M | 100 | | ✓ | 87.5 | 71.9 | 68.5 | 76.0 |
| VECO (Luo et al., 2020) | 662M | 662M | 50 | | ✓ | 87.0 | 70.4 | 68.0 | 75.1 |
| *Models fine-tuned only on English task data* | | | | | | | | | |
| XLM-R (Conneau et al., 2020a) | 559M | 559M | 100 | | | 82.8 | 69.0 | 62.3 | 71.4 |
| RemBERT (ours) | 995M | 575M | 110 | | | **84.2** | **73.3** | **68.6** | **75.4** |

additional task or translation data for fine-tuning, which significantly boosts performance (Hu et al., 2020). XLM-R and RemBERT are the only two models that are fine-tuned using only the English training data of the corresponding task. XLM-R was trained with a batch size of $2^{13}$ sequences each with $2^9$ tokens and 1.5M steps (total of 6.3T tokens). In comparison, RemBERT is trained with $2^{11}$ sequences of $2^9$ tokens for 1.76M steps (1.8T tokens). Even though it was trained with $3.5\times$ fewer tokens and has 10 more languages competing for the model capacity, RemBERT outperforms XLM-R on all tasks we considered. This strong result suggests that our proposed methods are also effective at scale. We will release the pre-trained model checkpoint and the source code for RemBERT in order to promote reproducibility and share the pre-training cost with other researchers.

# 6 ON THE IMPORTANCE OF THE OUTPUT EMBEDDING SIZE

We carefully design a set of experiments to analyze the impact of an increased output embedding size on various parts of the model. We study the nature of the decoupled input and output representations (§6.1) and the transferability of the Transformer layers with regard to task-specific (§6.2) and language-specific knowledge (§6.3).

## 6.1 NATURE OF INPUT AND OUTPUT EMBEDDING REPRESENTATIONS

We first investigate to what extent the representations of decoupled input and output embeddings differ based on word embedding association tests (Caliskan et al., 2017). Similar to Press & Wolf (2017), for a given pair of words, we evaluate the correlation between human similarity judgements of the strength of the relationship and the dot product of the word embeddings. We evaluate on MEN (Bruni et al., 2014), MTurk771 (Halawi et al., 2012), Rare-Word (Luong et al., 2013), SimLex999 (Hill et al., 2015), and Verb-143 (Baker et al., 2014). As our model uses subwords, we average the token representations for words with multiple subwords.

We show the results in Table 8. In the first two rows, we can observe that the input embedding of the decoupled model performs similarly to the embeddings of the coupled model while the output

Table 8: Results on word embedding association tests for the input (I) and output (O) embeddings of models (left) and the models' masked language modeling performance (right). The first two rows show the performance of coupled and decoupled embeddings with the same embedding size $E_{\text{in}} = E_{\text{out}} = 768$. The last three rows show the performance as we increase the output embedding size with $E_{\text{in}} = 128$.

| | MEN | | MTurk771 | | Rare-Word | | Simlex999 | | Verb-143 | | | MLM acc. |
|---|---|---|---|---|---|---|---|---|---|---|---|---|
| | I | O | I | O | I | O | I | O | I | O | | |
| Coupled | 40.8 | | 37.5 | | 25.0 | | 20.1 | | 56.0 | | Coupled | 61.1 |
| Decoupled | 39.2 | 27.7 | 37.5 | 24.3 | 24.0 | 12.2 | 17.6 | 16.1 | 59.4 | 43.9 | Decoupled | 61.6 |
| $E_{\text{out}} = 128$ | 40.7 | 36.6 | 37.7 | 32.8 | 23.6 | 16.4 | 17.5 | 17.3 | 48.9 | 46.4 | $E_{\text{out}} = 128$ | 59.0 |
| $E_{\text{out}} = 768$ | 38.6 | 27.8 | 35.2 | 23.9 | 22.6 | 11.5 | 19.7 | 15.6 | 50.6 | 45.5 | $E_{\text{out}} = 768$ | 60.7 |
| $E_{\text{out}} = 3072$ | 40.1 | 10.8 | 36.2 | 8.8 | 22.6 | -1.2 | 18.9 | 13.0 | 43.3 | 19.5 | $E_{\text{out}} = 3072$ | 62.3 |

embeddings have lower scores.[7] We note that higher scores are not necessarily desirable as they only measure how well the embedding captures semantic similarity at the lexical level. Focusing on the *difference* in scores, we can observe that the input embedding learns representations that capture semantic similarity in contrast to the decoupled output embedding. At the same time, the decoupled model achieves higher performance in masked language modeling.

The last three rows of Table 8 show that as $E_{\text{out}}$ increases, the difference in the input and output embedding increases as well. With additional capacity, the output embedding progressively learns representations that differ more significantly from the input embedding. We also observe that the MLM accuracy increases with $E_{\text{out}}$. Collectively, the results in Table 8 suggest that with increased capacity, the output embeddings learn representations that are worse at capturing traditional semantic similarity (which is purely restricted to the lexical level) while being more specialized to the MLM task (which requires more contextual representations). Decoupling embeddings thus give the model the flexibility to avoid encoding relationships in its output embeddings that may not be useful for its pre-training task. As pre-training performance correlates well with downstream performance (Devlin et al., 2019), forcing output embeddings to encode lexical information can hurt the latter.

## 6.2 CROSS-TASK TRANSFERABILITY OF TRANSFORMER LAYER REPRESENTATIONS

We investigate to what extent more capacity in the output embeddings during pre-training reduces the MLM-specific burden on the Transformer layers and hence prevents them from over-specializing to the MLM task.

**Dropping the last few layers** We first study the impact of an increased output embedding size on the transferability of the last few layers. Previous work (Zhang et al., 2020; Tamkin et al., 2020) randomly reinitialized the last few layers to investigate their transferability. However, those parameters are still present during fine-tuning. We propose a more aggressive pruning scheme where we completely remove the last few layers. This setting demonstrates more drastically whether a model's upper layers are over-specialized to the pre-training task by assessing whether performance can be improved with millions fewer parameters.[8]

We show the performance of models with 8–12 remaining layers (removing up to 4 of the last layers) for different output embedding sizes $E_{\text{out}}$ on XNLI in Figure 1. For both $E_{\text{out}} = 128$ and $E_{\text{out}} = 768$, removing the last layer improves performance. In other words, the model performs better even with 7.1M fewer parameters. With $E_{\text{out}} = 128$, the performance remains similar when removing the last few layers, which suggests that the last few layers are not critical for transferability.

As we increase $E_{\text{out}}$, the last layers become more transferable. With $E_{\text{out}} = 768$, removing more than one layer results in a sharp reduction in performance. Finally when $E_{\text{out}} = 3072$, every layer is

---

[7]This is opposite from what Press & Wolf (2017) observed in 2-layer LSTMs. They find that performance of the output embedding is similar to the embedding of a coupled model. This difference is plausible as the information encoded in large Transformers changes significantly throughout the model (Tenney et al., 2019).

[8]Each Transformer layer with $H = 768$ has about 7.1M parameters.

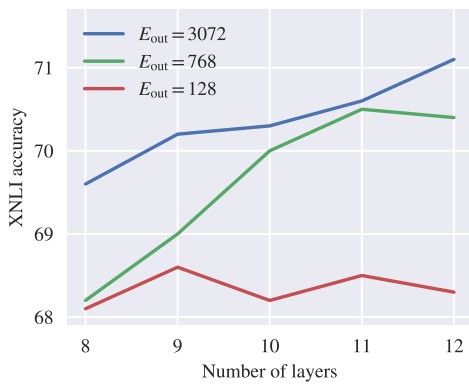
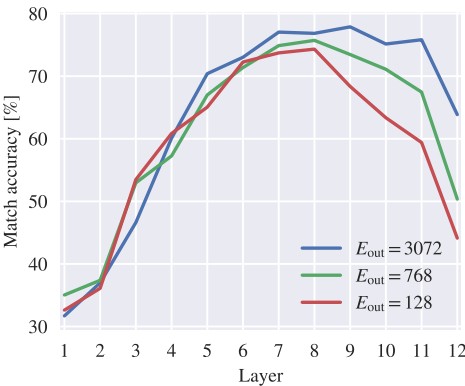

Figure 1: XNLI accuracy with the last layers removed. Larger $E_{\text{out}}$ improves transferability.

Figure 2: Nearest-neighbor English-to-German translation accuracy of each layer.

Table 9: Probing analysis of Tenney et al. (2019) with `mix` strategy.

| | # PT params | # FT params | POS | Const. | Deps. | Entities | SRL | Coref. O | Coref. W | SPR1 | SPR2 | Rel. | Avg |
|---|---|---|---|---|---|---|---|---|---|---|---|---|---|
| $E_{\text{out}} = 128$ | 115M | 100M | 96.7 | 87.9 | 94.3 | 93.7 | 91.7 | 95.0 | 67.2 | 83.0 | 82.7 | 77.0 | 86.9 |
| $E_{\text{out}} = 768$ | 192M | 100M | 96.7 | 87.9 | 94.4 | 94.0 | 91.8 | 95.0 | 67.0 | 83.1 | **82.8** | 78.6 | 87.1 |
| $E_{\text{out}} = 3072$ | 469M | 100M | **96.8** | **88.0** | **94.5** | 94.2 | **92.0** | **95.3** | **67.6** | **84.1** | 82.6 | **78.9** | **87.4** |

useful and removing any layer worsens the performance. This analysis demonstrates that increasing $E_{\text{out}}$ improves the transferability of the representations learned by the last few Transformer layers.

**Probing analysis**   We further study whether an increased output embedding size improves the general natural language processing ability of the Transformer. We employ the probing analysis of Tenney et al. (2019) and the `mix` probing strategy where a 2-layer dense network is trained on top of a weighted combination of the 12 Transformer layers. We evaluate performance with regard to core NLP concepts including part-of-speech tagging (POS), constituents (Consts.), dependencies (Deps.), entities, semantic role labeling (SRL), coreference (Coref.), semantic proto-roles (SPR), and relations (Rel.). For a thorough description of the task setup, see Tenney et al. (2019).[9]

We show the results of the probing analysis in Table 9. As we increase $E_{\text{out}}$, the model improves across all tasks, even though the number of parameters is the same. This demonstrates that increasing $E_{\text{out}}$ enables the Transformer layers to learn more general representations.[10]

### 6.3   CROSS-LINGUAL TRANSFERABILITY OF TRANSFORMER LAYER REPRESENTATIONS

So far, our analyses were not specialized to multilingual models. Unlike monolingual models, multilingual models have another dimension of transferability: cross-lingual transfer, the ability to transfer knowledge from one language to another.

Previous work (Pires et al., 2019; Artetxe et al., 2020) has found that MLM on multilingual data encourages cross-lingual alignment of representations without explicit cross-lingual supervision. While it has been shown that multilingual models learn useful cross-lingual representations, over-specialization to the pre-training task may result in higher layers being less cross-lingual and focusing on language-specific phenomena necessary for predicting the next word in a given language. To investigate to what extent this is the case and whether increasing $E_{\text{out}}$ improves cross-lingual alignment, we evaluate the model's nearest neighbour translation accuracy (Pires et al., 2019) on English-to-German translation (see Appendix A.9 for a description of the method).

---

[9]The probing tasks are in English while our encoder is multilingual.

[10]In Tenney et al. (2019), going from a BERT-base to a BERT-large model (with $3\times$ more parameters) improves performance on average by 1.1 points, compared to our improvement of 0.5 points without increasing the number of fine-tuning parameters.

We show the nearest neighbor translation accuracy for each layer in Figure 2. As $E_{\text{out}}$ increases, we observe that a) the Transformer layers become more language-agnostic as evidenced by higher accuracy and b) the language-agnostic representation is maintained to a higher layer as indicated by a flatter slope from layer 7 to 11. In all cases, the last layer is less language-agnostic than the previous one. The sharp drop in performance after layer 8 at $E_{\text{out}} = 128$ is in line with previous results on cross-lingual retrieval (Pires et al., 2019; Hu et al., 2020) and is partially mitigated by an increased $E_{\text{out}}$. In sum, not only does a larger output embedding size improve cross-task transferability but it also helps with cross-lingual alignment and thereby cross-lingual transfer on downstream tasks.

## 7 Conclusion

We have assessed the impact of embedding coupling in pre-trained language models. We have identified the main benefit of decoupled embeddings to be the flexibility endowed by decoupling their shapes. We showed that input embeddings can be safely reduced and that larger output embeddings and reinvesting saved parameters lead to performance improvements. Our rebalanced multilingual BERT (RemBERT) outperforms XLM-R with the same number of fine-tuning parameters while having been trained on $3.5\times$ fewer tokens. Overall, we found that larger output embeddings lead to more transferable and more general representations, particularly in a Transformer's upper layers.

## Acknowledgements

We would like to thank Laura Rimell for valuable feedback on a draft of this paper.

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

## A  APPENDIX

### A.1  EFFICIENCY COMPARISON BASED ON PARAMETER COUNT DURING FINE-TUNING

We compare the efficiency of models based on their number of parameters. We believe this to be a reasonable proxy for a model's efficiency as the performance of Transformer-based language models has been shown to improve monotonically with the number of parameters (Kaplan et al., 2020; Raffel et al., 2020; Lepikhin et al., 2020; Brown et al., 2020; Shoeybi et al., 2019; Aharoni et al., 2019). As the number of parameters during pre-training and fine-tuning may differ[11], we compare models based on their number of parameters during the fine-tuning stage (without the task-specific head). We argue that this is the most practically relevant number as a model is generally pre-trained only once but may be fine-tuned or used for inference millions of times.

---

[11]For encoder-only models such as BERT, parameters after the last Transformer layer (e.g. the output embeddings and the pooling layer) are discarded after pre-training.

Table 10: Fine-tuning hyperparameters for all models except RemBERT.

|        | Learning rate | Batch size | Train epochs |
|--------|---------------|------------|--------------|
| PAWS-X | $[3 \times 10^{-5},\ 4 \times 10^{-5},\ 5 \times 10^{-5}]$ | 32 | 3 |
| XNLI   | $[1 \times 10^{-5},\ 2 \times 10^{-5},\ 3 \times 10^{-5}]$ | 32 | 3 |
| SQuAD  | $[2 \times 10^{-5},\ 3 \times 10^{-5},\ 4 \times 10^{-5}]$ | 32 | 3 |
| NER    | $[1 \times 10^{-5},\ 2 \times 10^{-5},\ 3 \times 10^{-5}, 4 \times 10^{-5}, 5 \times 10^{-5}]$ | 32 | 3 |

Table 11: Statistics for the datasets in XTREME, including the number of training, development, and test examples as well as the number of languages for each task.

| Task | Corpus | |Train| | |Dev| | |Test| | |Lang.| | Task | Metric | Domain |
|------|--------|--------|-------|--------|---------|------|--------|--------|
| Classification | XNLI | 392,702 | 2,490 | 5,010 | 15 | NLI | Acc. | Misc. |
|  | PAWS-X | 49,401 | 2,000 | 2,000 | 7 | Paraphrase | Acc. | Wiki / Quora |
| Structured prediction | POS | 21,253 | 3,974 | 47-20,436 | 33 | POS | F1 | Misc. |
|  | NER | 20,000 | 10,000 | 1,000-10,000 | 40 | NER | F1 | Wikipedia |
| QA | XQuAD | 87,599 | 34,726 | 1,190 | 11 | Span extraction | F1 / EM | Wikipedia |
|  | MLQA |  |  | 4,517–11,590 | 7 | Span extraction | F1 / EM | Wikipedia |
|  | TyDiQA-GoldP | 3,696 | 634 | 323–2,719 | 9 | Span extraction | F1 / EM | Wikipedia |
| Retrieval | BUCC | - | - | 1,896–14,330 | 5 | Retrieval | F1 | Wiki / news |
|  | Tatoeba | - | - | 1,000 | 33 | Retrieval | Acc. | misc. |

## A.2 BASELINE MODEL DETAILS

Our baseline model has the same architecture as multilingual BERT (mBERT; Devlin et al., 2019). It consists of 12 Transformer layers with a hidden size $H$ of 768 and 12 attention heads with 64 dimensions each. Input and output embeddings are coupled and have the same dimensionality $E$ as the hidden size, i.e. $E_{\text{out}} = E_{\text{in}} = H$. The total number of parameters during pre-training and fine-tuning is 177M. We do not use dropout following the recommendation from Lan et al. (2020). We use the SentencePiece tokenizer (Kudo & Richardson, 2018) and a shared vocabulary of 120k subwords. The model is trained on Wikipedia dumps in 104 languages following Devlin et al. (2019) using masked language modeling (MLM). We choose this baseline as its behavior has been thoroughly studied (K et al., 2020; Conneau et al., 2020b; Pires et al., 2019; Wu & Dredze, 2019).

## A.3 TRAINING DETAILS

For all pre-training except for the large scale RemBERT, we trained using 64 Google Cloud TPUs. We trained over 26B tokens of Wikipedia data. All fine-tuning experiments were run on 8 Cloud TPUs. For all fine-tuning experiments other than RemBERT, we use batch size of 32. We sweep over the learning rate values specified in Table 10.

We used the SentencePiece tokenizer trained with unigram language modeling

## A.4 XTREME TASKS

For our experiments, we employ tasks from the XTREME benchmark (Hu et al., 2020). We show statistics for them in Table 11. XTREME includes the following datasets: The Cross-lingual Natural Language Inference (XNLI; Conneau et al., 2018) corpus, the Cross-lingual Paraphrase Adversaries from Word Scrambling (PAWS-X; Yang et al., 2019) dataset, part-of-speech (POS) tagging data from the Universal Dependencies v2.5 (Nivre et al., 2018) treebanks, the Wikiann (Pan et al., 2017) dataset for named entity recognition (NER), the Cross-lingual Question Answering Dataset (XQuAD; Artetxe et al., 2020), the Multilingual Question Answering (MLQA; Lewis et al., 2020) dataset, the gold passage version of the Typologically Diverse Question Answering (TyDiQA; Clark et al., 2020a) dataset, data from the third shared task of the workshop on Building and Using Parallel Corpora (BUCC; Zweigenbaum et al., 2018), and the Tatoeba dataset (Artetxe & Schwenk, 2019). We refer the reader to Hu et al. (2020) for more details. We average results across three fine-tuning runs and evaluate on the dev sets unless otherwise stated.

Table 12: Effect of reducing the embedding size $E$ for monolingual vs. multilingual models on MNLI and XNLI performance respectively. Monolingual numbers are from Lan et al. (2020) and have vocabulary size of 30k.

| English | # PT params | # FT params | MNLI | Multilingual | # PT params | # FT params | XNLI |
|---|---|---|---|---|---|---|---|
| $E = H = 768$ | 110M | 110M | 84.5 | $E = H = 768$ | 177M | 177M | 70.7 |
| $E = H = 128$ | 89M | 89M | 83.7 | $E = H = 128$ | 100M | 100M | 68.1 |

Table 13: Effect of an increased output embedding size $E_{\text{out}}$ and additional layers during pre-training $L = 15$ on English BERT$_{\text{Base}}$ ($E_{\text{in}} = 128$).

| | # PT params | # FT params | MNLI Acc | SQuAD EM/F1 |
|---|---|---|---|---|
| BERT$_{\text{Base}}$ (ours) | 110M | 110M | 79.8 | 78.4/86.2 |
| $E_{\text{out}} = 128$ | 93M | 89M | 75.9 | 75.5/84.2 |
| $E_{\text{out}} = 768$ | 112M | 89M | 77.5 | 77.5/85.5 |
| $E_{\text{out}} = 3072$ | 181M | 89M | 79.5 | 78.4/86.2 |
| $L = 15$ | 114M | 89M | 80.1 | 78.7/86.3 |
| $L = 24$ | 178M | 89M | 79.0 | 77.8/85.5 |

## A.5 COMPARISON TO LAN ET AL. (2020)

Crucially, our finding differs from the dimensionality reduction in ALBERT (Lan et al., 2020). While they show that smaller embeddings can be used, their input and output embeddings are coupled and use a much smaller vocabulary (30k vs 120k). In contrast, we find that simultaneously decreasing both the input and output embedding size drastically reduces the performance of multilingual models.

In Table 12, we show the impact of their factorized embedding parameterization on a monolingual and a multilingual model. While the English model suffers a smaller (0.8%) drop in accuracy, the multilingual model's performance drops by 2.6%. Direct application of a factorized embedding parameterization (Lan et al., 2020) is thus not viable for multilingual models.

## A.6 ENGLISH MONOLINGUAL RESULTS

So far, we have focused on multilingual models as the number of saved parameters when reducing the input embedding size is largest for them. We now apply the same techniques to the English 12-layer BERT$_{\text{Base}}$ with a 30k vocabulary (Devlin et al., 2019). Specifically, we decouple the embeddings, reduce $E_{\text{in}}$ to 128, and increase the output embedding size or the number of layers during pre-training. We show the performance on MNLI (Williams et al., 2018) and SQuAD (Rajpurkar et al., 2016) in Table 13. By adding more capacity during pre-training, performance monotonically increases similar to the multilingual models. Interestingly, pruning a 24-layer model to 12 layers reduces performance, presumably because some upper layers still contain useful information.

## A.7 REMBERT DETAILS

We design a Rebalanced mBERT (RemBERT) to leverage capacity more effectively during pre-training. The model has 995M parameters during pre-training and 575M parameters during fine-tuning. We pre-train on large unlabeled text using both Wikipedia and Common Crawl data, covering 110 languages. The details of hyperparameters and architecture are shown in Table 14.

For each language $l$, we define the empirical distribution as

$$p_l = \frac{n_l}{\sum_{l' \in L} n_{l'}} \tag{1}$$

Table 14: Hyperparameters for RemBERT architecture and pre-training.

| Hyperparameter | RemBERT |
|---|---|
| Number of layers | 32 |
| Hidden size | 1152 |
| Vocabulary size | 250,000 |
| Input embedding dimension | 256 |
| Output embedding dimension | 1536 |
| Number of attention heads | 18 |
| Attention head dimension | 64 |
| Dropout | 0 |
| Learning rate | 0.0002 |
| Batch size | 2048 |
| Train steps | 1.76M |
| Adam $\beta_1$ | 0.9 |
| Adam $\beta_2$ | 0.999 |
| Adam $\epsilon$ | $10^{-6}$ |
| Weight decay | 0.01 |
| Gradient clipping norm | 1 |
| Warmup steps | 15000 |

Table 15: Hyperparameters for RemBERT fine-tuning.

| | Learning rate | Batch size | Train epochs |
|---|---|---|---|
| PAWS-X | $8 \times 10^{-6}$ | 128 | 3 |
| XNLI | $1 \times 10^{-5}$ | 128 | 3 |
| SQuAD | $9 \times 10^{-6}$ | 128 | 3 |
| POS | $3 \times 10^{-5}$ | 128 | 3 |
| NER | $8 \times 10^{-6}$ | 64 | 3 |

where $n_l$ is the number of sentences in $l$'s pre-training corpus. Following Devlin et al. (2019), we use an exponentially smoothed distribution, i.e., we exponentiaate $p_l$ by $\alpha = 0.5$ and renormalize to obtain the sampling distribution.

Hyperparameters and pre-training details are summarized in Table 14. Hyperparameters used for the leaderboard submission are shown in Table 15.

### A.8 XTREME TASK RESULTS

We show the detailed results for RemBERT and the comparison per task on the XTREME leaderboard in Table 16. Compared to Table 7, which shows the average across task categories, this table shows the average across tasks.

### A.9 NEAREST-NEIGHBOR TRANSLATION COMPUTATION

For an English-to-German translation, we sample $M = 5000$ pairs of sentences from WMT16 (Bojar et al., 2016). For each sentence in each language, we obtain a representation $v_{\text{LANG}}^{(l)}$ at each layer $l$ by averaging the activations of all tokens (except the [CLS] and [SEP] tokens) at that layer. We then compute a translation vector from English to German by averaging the difference between the vectors of each sentence pair across all pairs: $\bar{v}_{\text{EN}\rightarrow\text{DE}}^{(l)} = \frac{1}{M} \sum_{i=1}^{M} \left( v_{\text{DE}_i}^{(l)} - v_{\text{EN}_i}^{(l)} \right)$.

For each English sentence $v_{\text{EN}_i}^{(l)}$, we can now translate it with this vector: $v_{\text{EN}_i}^{(l)} + \bar{v}_{\text{EN}\rightarrow\text{DE}}^{(l)}$. We locate the closest German sentence vector based on $\ell_2$ distance and measure how often the nearest neighbour is the correct pair.

Table 16: Comparison of our model to other models on the XTREME leaderboard. Details about VECO are due to communication with the authors. Avg$_{\text{task}}$ is averaged over tasks whereas Avg is averaged over task categories just like Table 7.

| | # PT params | # FT params | XNLI Acc | POS F1 | NER F1 | PAWS-X Acc | XQuAD EM/F1 | MLQA EM/F1 | TyDi-GoldP EM/F1 | Avg$_{\text{task}}$ | Avg |
|---|---|---|---|---|---|---|---|---|---|---|---|
| *Models fine-tuned on translations or additional task data* | | | | | | | | | | | |
| STiLTs (Phang et al., 2020) | 559M | 559M | 80.0 | 74.9 | 64.0 | 87.9 | 63.3/78.7 | 53.7/72.4 | 59.5/76.0 | 72.7 | 73.5 |
| FILTER (Fang et al., 2020) | 559M | 559M | 83.9 | 76.2 | 67.7 | 91.4 | 68.0/82.4 | 57.7/76.2 | 50.9/68.3 | 74.9 | 76.0 |
| VECO (Luo et al., 2020) | 662M | 662M | 83.0 | 75.1 | 65.7 | 91.1 | 66.3/79.9 | 54.9/73.1 | 58.9/75.0 | 74.1 | 75.1 |
| *Models fine-tuned only on English task data* | | | | | | | | | | | |
| XLM-R (Conneau et al., 2020a) | 559M | 559M | 79.2 | 73.8 | 65.4 | 86.4 | 60.8/76.6 | 53.2/71.6 | 45.0/65.1 | 70.1 | 71.4 |
| RemBERT (ours) | 995M | 575M | **80.8** | **76.5** | **70.1** | **87.5** | **64.0/79.6** | **55.0/73.1** | **63.0/77.0** | **74.4** | **75.4** |

