# OpenReview forum: "Rethinking Embedding Coupling in Pre-trained Language Models"
_ICLR.cc/2021/Conference — ICLR 2021 Poster_

### Official Review · AnonReviewer4 · 2020-10-21
**much larger model size?**

**Rating:** 4
**Confidence:** 5

**Review:**

This work studied the impact of embedding coupling in pre-trained language models, by taking a multilingual model as backbone. The major finding is that decoupling the input and output embedding shapes can bring benefits, and the output embedding plays an important role in the transferability of pre-trained representation. A rebalanced mBERT is designed by combing and scaling up the investigated techniques, achieving strong results on the XTREME benchmark.

This paper is well written, and the proposed strategy is simple yet effective for obtaining more transferable language representations. The insights of model design for more efficient fine-tuning are well supported by the analysis.

My concern is about the model efficiency and the true source of performance improvement. It seems that the number of the parameters is much more than those public ones. I am curious if it is a fair comparison in Table 7.

The current approach does help in reducing the number of parameters in the fine-tuning stage by increasing E_{out}. Even in your all experiments, the optimal value of E_{out} appears to be 768 (similar in the baseline), where performance is quite similar to baseline. However, increasing E_{out} to a much larger value (i.e 3072) drastically increases the number of parameters in pre-training, even as compared to baseline, which seems like a trade-off between fine-tuning and pre-training. The performance with much larger E_{out} is a marginal improvement over baseline unless the saved parameters are reinvested. The reinvestment of parameters creates a larger improvement over baseline (Table 6). The authors must conclude with optimal values of E_{in} and E_{out}, otherwise the paper is merely a series of experiments with different values of E_{in}, E_{out}, and # of layers in the baseline.

Try referencing Table 7 in the main explanation of RemBERT rather than in the Appendix, as it is your major contribution. But again my concern is the same, the number of parameters in fine-tuning is larger than XLM-R plus the additional pre-training time (more than XLM-R). Yes, the performance increases but the main objective of the paper is not quite satisfied (reducing the number of parameters in FT which are actually larger in RemBERT than XML-R). Maybe you can alter the abstract accordingly and focus on reinvestment, which actually is helping in making your point in the experiments.



Minor comments:

Check the use of \citep{} and \citet{}, e.g., in the last sentence of Section 2 and Footnote 10.

Table 7 as main results was never cited in this paper (but only in appendix), which is not a proper organization way.

Page 3-> "It consists of 12 Transformer layers with a hidden size H of 768 and 12" the sentence seems incomplete.

Page 5 -> "increasing the number of Transformer layers L results in the best performance" I am wondering if reinvestment in L gains higher performance than in H because of the larger number of parameters (10M more). Maybe you can keep reinvestment the same in both H and L for fair comparison and better results projection.

Section 4: How about a proportional increase in E_{in} and E_{out} affects the performance (e.g increasing or decreasing both with the same proportion, especially decreasing which lines up with paper goal i.e providing flexibility with less number of parameters.)

Page 7 -> "For both E_{out} = 128 and E_{out} = 768, removing the last layer improves performance". Seems in disagree with Figure 1. Acc with 10 layers is lower than that with 12 layers for E_{out}=128. Similar is the case with E_{out}=768.

How about the extra pre-training and fine-tuning time compared with the baseline?

For the final rebalanced mBERT described in page 5, how are the hyper-parameters decided?

---

> ### Author Response · Authors · 2020-11-20
> **We included additional data to back up our claim that the comparison between RemBERT and XLM-R is fair. We also clarified the main objective of the paper.**
>
> > “It seems that the number of the parameters is much more than those public ones. I am curious if it is a fair comparison in Table 7.”
>
> We reemphasize that we constructed RemBERT to have a number of fine-tuning parameters that matches XLM-R’s as closely as possible while making the most effective use of our techniques and the accelerators that were available to us. Since we used Google Cloud TPUs, tensor dimensions that are multiples of 128 are more efficient. The resulting model has 575M vs XLM-R’s 559M parameters, so only around 16M (or 2.9%) more parameters. We respectfully disagree with the characterization that 2.9% more parameters are "much more". We think the parameter sizes are similar enough for such large models to ensure a fair comparison. In order to back up our claim, we ran the following experiment. We removed the last 2 layers (each layer has 16M parameters) of RemBERT so that the total number of parameters is now 16M fewer than that of XLM-R. The XNLI accuracy is only reduced from 80.8 to 80.6. Considering that RemBERT has a 4% higher score averaged over 7 tasks, a drop of 0.2% is small. This demonstrates that the effect of a 32M parameters for these large models is not significant and doesn’t change our conclusion that RemBERT outperforms XLM-R.
>
> Additionally, as mentioned in the paper, RemBERT is trained with 3.5X fewer tokens and covers 10 more languages than XLM-R. Therefore, per-language capacity of RemBERT (5.23M params/lang) is lower for RemBERT than that of XLM-R (5.59M params/lang).
>
> Finally, we found that the pre-training time for XLM-R is 52% more time than that of RemBERT. For details on how we got these numbers, please refer to our second response.
>
> With these factors in mind, we believe that our comparison turns out to be unfair for RemBERT and our claim that RemBERT outperforms XLM-R is a very conservative one.
>
> > “The current approach does help in reducing the number of parameters in the fine-tuning stage by increasing E_{out}. … merely a series of experiments with different values of E_{in}, E_{out}, and # of layers in the baseline.”
>
> We believe that this paragraph is based on a comparison between models with different sizes, which we believe is not fair. We try our best to vary only one variable in a table such that the effect of that variable can be studied in isolation. Since models in Table 4 have 100M parameters during fine-tuning, comparing them to the baseline with 177M parameters (43% more) mixes the effects of the number of parameters and larger embedding size. As such, we can’t draw conclusions from such a comparison.
>
> The purpose of Table 4 is to show that given the same number of fine-tuning parameters, increasing the output embedding size improves the performance and not that increasing the output embedding at much smaller model size can outperform the baseline.
>
> Therefore, we respectfully argue that the statement that “The performance with much larger E_{out} is a marginal improvement over baseline unless the saved parameters are reinvested” misses the main purpose of Table 4. We will make it clear in Table 4 that they should not be compared to the baseline.
>
> > “seems like a trade-off between fine-tuning and pre-training”
>
> We would like to highlight that our paper—to our knowledge—is the first one that shows that there exists such a trade-off between the number of pre-training and fine-tuning parameters. This is indeed a trade-off between exchanging slightly increased pre-training computation for improved downstream performance.
>
> Typically a model is used for inference many times whereas it is trained very few times as the cost of training is not justified otherwise. Furthermore, well-trained models are often publicly released so that they can be used by many users for fine-tuning and inference. In such scenarios, the cost of training is amortized over a number of people. Therefore, we believe that the cost of additional compute that can be shared between multiple inference runs and people is justified. But little or no previous work has been done on studying such trade-offs between pre-training and fine-tuning. We believe that our paper fills the gap by exploring various ways of adding parameters only during pre-training.
>
> > “Table 7 [...] is your major contribution”
>
> We respectfully disagree. While Table 7 demonstrates that our methods work at scale and can be used to outperform the state of the art with similar model size, our main contributions are the effects of embedding decoupling, the importance of the output embedding size, and analyses with regard to their impact on model behavior.
>
> > “the optimal value of E_{out} appears to be 768 (similar in the baseline)”
>
> We think there might be some misunderstanding here. In Table 4, we show that $E_{out} = 3072$ performs even better than $E_{out} = 768$.

---

> > ### Author Response · Authors · 2020-11-20
> > **continued response**
> >
> > > But again my concern is the same, the number of parameters in fine-tuning is larger than XLM-R plus the additional pre-training time (more than XLM-R).
> >
> > In the paper, we mentioned that RemBERT was trained with 3.5x fewer tokens during pre-training compared to XLM-R. However, RemBERT has more parameters during pre-training. We believe that these factors lead the reviewer to think that RemBERT requires additional pre-training time.
> >
> > In order to make a fair comparison, we profiled the XLM-R and RemBERT using exactly the same hardware setting. We ran XLM-R for 10000 steps just to be able to profile reliably and did not complete the training. In addition, for XLM-R, we used the exact training setting from the paper (Conneau et al. 2020). XLM-R was trained with 4 times larger batch size (8192 vs. 2048), which significantly increases the training time. Therefore, even though the RemBERT model had more pre-training parameters, each gradient update step was faster; it took 0.348 sec. for RemBERT whereas XLM-R takes 0.6208 sec. The total estimated pre-training time is listed in the table below. XLM-R pre-training takes 52% more time than RemBERT.
> >
> > || Pre-training time|
> > |:---  |:---:|
> > |XLM-R|259 hr|
> > |RemBERT|170 hr|
> >
> > Overall, these profiling results suggest that RemBERT is faster to train and was able to outperform XLM-R with much less resources during pre-training.
> >
> >
> > > Yes, the performance increases but the main objective of the paper is not quite satisfied (reducing the number of parameters in FT which are actually larger in RemBERT than XML-R). Maybe you can alter the abstract accordingly and focus on reinvestment, which actually is helping in making your point in the experiments.
> >
> > We would like to clarify that the main objective of the paper is not to reduce the number of FT parameters. Our main objective was to study the practice of embedding coupling and make parameter allocation in pre-trained language models more efficient overall and representations more generalizable. If there are specific parts of the paper that suggest that our objective is to reduce the FT parameters, please let us know. We will modify the text to reduce the source of confusion.
> >
> >
> > The remaining part of the response addresses stylistic feedback and minor comments.
> >
> > > Check the use of \citep{} and \citet{}.
> >
> > We fixed incorrect uses of \citep in the main text and the Appendix.
> >
> > > Table 7 as main results was never cited in this paper (but only in appendix).
> >
> > We have fixed the link to Table 7.
> >
> > > Page 3-> "It consists of 12 Transformer layers with a hidden size H of 768 and 12" the sentence seems incomplete.
> >
> > Thanks for catching this. We fixed this in the revised text.
> >
> > > Page 5 -> "increasing the number of Transformer layers L results in the best performance" I am wondering if reinvestment in L gains higher performance than in H because of the larger number of parameters (10M more).
> >
> > In our experiments, 10M more parameters (or 5.6% more) do not result in a performance difference of 1.7, which is the difference in the average score between the ”Reinvested in $H$” and “reinvested in $L$” models in Table 6. We tried removing 2 layers of the “Reinvested in $L$” model so that the total number of parameters is about 164M, which is 4M less than the “Reinvested in $H$”. The performance was almost identical to the original model without the 2 layers removed.
> >
> > > Section 4: How about a proportional increase in E_{in} and E_{out} affects the performance (e.g increasing or decreasing both with the same proportion, especially decreasing which lines up with paper goal i.e providing flexibility with less number of parameters.)
> >
> > Again, we want to argue that our goal is not reducing the number of parameters.
> >
> > > Page 7 -> "For both E_{out} = 128 and E_{out} = 768, removing the last layer improves performance". Seems in disagree with Figure 1. Acc with 10 layers is lower than that with 12 layers for E_{out}=128. Similar is the case with E_{out}=768.
> >
> > We would like to clarify that “removing the last layer” refers to the data point with 11 layers, not 10 (as the full model has 12 layers). In Figure 1, we indeed see a slight bump in performance with 11 layers for $E_{\rm{out}} =128$ and $E_{\rm{out}} = 768$.
> >
> > > How about the extra pre-training and fine-tuning time compared with the baseline?
> >
> > Please refer to the table we included in the response to R1 where we discuss the training/inference time of our strategies.
> >
> > > For the final rebalanced mBERT described in page 5, how are the hyper-parameters decided?
> >
> > We built the RemBERT model combining various improvement strategies. For example, we added 8 extra layers because “Reinvested in $L$” performed the best. However, adding too many layers may result in instability typical of deep neural networks. Therefore, we also used the “Reinvested in $H$” recipe. All the dimensions are multiple of 128 to make sure that our model efficiently harnesses accelerators. The vocabulary size was chosen to be the same as XLM-R.

---

### Official Review · AnonReviewer3 · 2020-10-28
**A well-written paper and an exciting idea**

**Rating:** 6
**Confidence:** 3

**Review:**

This paper systematically studies the impact of embedding coupling with multilingual language models.  The authors observe that while na¨ıvely decoupling the input and output embedding parameters does not consistently improve downstream evaluation metrics, decoupling their shapes comes with a host of benefits. Moreover, they achieve significantly improved performance by reinvesting saved parameters to the width and depth of the Transformer layers on the XTREME benchmark.

This paper is well-written and strongly motivated. The idea of decoupling embedding is novel, and the evaluation results are strong.

Strength:

+ The systematical study of the impact of embedding coupling on state-of-the-art pre-trained language models.  This paper also thoroughly investigates reasons for the benefits of embedding decoupling and observes that an increased output embedding size enables a model to improve on the pre-training task, which correlates with downstream performance. They also find that it leads to Transformers that are more transferable across tasks and languages. Those empirical results will promote future model design and the understanding of the transformer for textual representation learning.
+ The paper proposes a method to reinvest saved parameters to the width and depth of the Transformer layers and achieve significantly improved performance on the XTREME benchmark over a strong mBERT.

Weakness:

- Some model details are missing. Although I know what the input and output embedding are, it is still a bit hard to follow in Section 4 and 5. I strongly recommend that the authors revise those parts and introduce the model details, such as decoupling and model architecture in your experiments.
- Lots of your model designs are empirical such as the embedding size, and it is a bit boring to optimize those hyperparameters, and sometimes we even do not know why it works.

Questions:

- Could you please introduce the model details of your decoupled model in Section 4?

---

> ### Author Response · Authors · 2020-11-19
> **Added more details about the model architecture throughout the paper.**
>
> > “Some model details are missing. Although I know what the input and output embedding are, it is still a bit hard to follow in Section 4 and 5.”
>
> We added further details on the Transformer architecture and the decoupling information in the text and table captions not just in Sections 4 and 5 but throughout the paper. Please let us know if there are other model details that you believe are missing.
>
> > “Lots of your model designs are empirical such as the embedding size, and it is a bit boring to optimize those hyperparameters, and sometimes we even do not know why it works.”
>
> We agree with the reviewer’s notion that improving models by tweaking hyperparameters such as embedding size can be effective but less interesting from a research point of view. However, given that one of the most important contributions of our paper happens to be increasing the output embedding size, we put significant emphasis (dedicate the entire section 6, which is also the longest section) on understanding why increasing the output embedding improves the performance. We found that larger output embedding size during pre-training improves the transferability to fine-tuning tasks and to other languages. In fact, R2 mentions this as one of the strong points (item 3) and R4 mentions that “The insights of model design for more efficient fine-tuning are well supported by the analysis.” For these reasons, we hope that the reviewer can appreciate our efforts in better understanding the profound impact decoupling and the choice of the output embedding size may have on pre-trained language models.
>
> Overall, our analysis illustrates that there are no universally optimal hyper-parameters. Instead, we highlight trade-offs that have so far been unexplored between additional compute only during pre-training (e.g., larger output embedding) and improved fine-tuning performance.
>
> Given the overall positive nature of the review and the only minor negative points raised, we would like to encourage the reviewer to potentially reconsider their score.

---

### Official Review · AnonReviewer1 · 2020-10-28
**Good Analysis**

**Rating:** 7
**Confidence:** 3

**Review:**

Summary:

This work investigated the strategy of reallocating parameters of multilingual language models for improving their cross lingual transferability. Authors first decoupled the input and output embeddings and showed that the capacity of output embedding is more important than input embedding. Then,  they proposed a Rebalanced mBERT (RemBERT) model that reallocates the input embedding parameters of mBERT model to the output embedding and additional layers. Experimental results on XTREME benchmark showed that RemBERT significantly outperformed XLM-R with similar model size.

Pros:

- The paper is well written and easy to follow. The comparison of embeddings parameters ratios of different language models in Table 1 gives a very good motivation of reallocating the parameters of embeddings.

- Authors conducted several ablation studies to understand how the capacity of  different parts of the model (e.g., input and output embeddings, transformer layers) contributes to the final performance.

Cons:

- Reallocating the parameters of input embedding to additional transformer layers might affect the pre-training and inference speed, it will be helpful to show the training and inference speed of different parameter reallocation strategies.

Questions:

- Why are the pre-training and fine-tuning parameters of coupled and decoupled models the same in Table 2?  Shouldn’t decoupling the input and output embeddings double the parameters of the embeddings?

---

> ### Author Response · Authors · 2020-11-19
> **We added pre-training and inference speed information.**
>
> > “Reallocating the parameters of input embedding to additional transformer layers might affect the pre-training and inference speed, it will be helpful to show the training and inference speed of different parameter reallocation strategies.”
>
> Thanks for the suggestion. We ran profiling for our reallocation strategies as well as larger output embedding experiments.
>
> |                                	        | Pre-training speed  [steps/s] 	|   # PT params   	|    # FT params 	|
> |:--------------------------------	| :--------------------:	| :-----------------:	|    :--------:|
> | $E_{\textrm{out}} = 128$       	|                         13.2 	|        115M           	|        100M 	|
> | $E_{\textrm{out}} = 768$       	|                         11.3 	|        192M     	|        100M 	|
> | $E_{\textrm{out}} = 3072$       |                          7.6 	|        469M            |        100M 	|
> | 11 add. layers (Table 5) 	         |                          7.2 	|        193M 	        |        100M 	|
>
> We show above the pre-training speed for the models in Tables 4 and 5. The fine-tuning speed should be the same as they share the same 12 layer architecture with the same hidden size. When the output embedding size is increased from 128 to 3072 (factor of 24 increase), the pre-training speed is reduced only by 42%. This is because the additional compute associated with larger output embedding is highly optimized via dense matrix multiplication.
>
>
>
> |                   	| Pre-training speed [steps/s] 	| Inference speed [s] 	| # PT params 	| # FT params 	|
> |-------------------	|-----------------------------:	|:-------------------:	|:-----------:	|:-----------:	|
> | Baseline          	|                         10.6 	|               21.7 	|        177M 	|        177M 	|
> | Reinvested in $H$ 	|                         7.8 	|                20.3 	|        260M 	|        168M 	|
> | Reinvested in $L$ 	|                         6.7 	|                24.7 	|        270M 	|        178M 	|
>
> This table shows the pre-training and inference speed for the reinvesting strategies in Table 6. During pre-training both reinvested models are slower than the baseline. This is expected because of the additional pre-training capacity in the output embedding. More parameters translate to a more expensive all-reduce operation. Comparing between “Reinvested in $H$” and “Reinvested in $L$”, the latter is slightly slower during pre-training because the forward and backward pass through the deeper model can’t be easily parallelized whereas the additional compute associated with the larger hidden size consists of highly optimized dense matrix multiplication.
>
> To measure the inference speed, we ran the inference (i.e., forward pass only) on the XNLI test set which has 112,350 examples with batch size of 256. Since XNLI has only 3 classes, the task specific computation is negligible. The measured time does not include setting up the input pipeline. We ran with 8 Google Cloud TPUs.
>
> We observe that the “Reinvested in $H$” is the fastest. The reduced input embedding makes the embedding lookup faster while the larger hidden size is efficiently computed with highly optimized matrix multiplication. The “Reinvested in $L$” is 12.3% slower than the baseline, mainly due to the additional layers.
>
> Since the performance of “Reinvested in $L$” was better than “Reinvested in H” and the baseline, there exists a tradeoff between inference speed and the fine-tuning performance as well. This tradeoff motivated our design of the full-scale RemBERT model, which reinvests in both $H$ and $L$ in an attempt to strike the right balance.
>
> We note that the inference speed depends on the hardware and software. For example, some hardware can be optimized for the embedding lookup operation, in which case the baseline model may perform better than the “Reinvested in $H$” model.
>
>
> > Why are the pre-training and fine-tuning parameters of coupled and decoupled models the same in Table 2? Shouldn’t decoupling the input and output embeddings double the parameters of the embeddings?
>
> Yes you are correct. This is a typo. We fixed the number of parameters in Table 2. Thanks for catching this.

---

### Official Review · AnonReviewer2 · 2020-10-31
**Transformer based bidirectional LMs pre-trained using Masked Language Model loss typically share input and output token embeddings. This paper makes an interesting investigation about decoupling input and output embeddings and gains which can be obtained out of this decoupling.**

**Rating:** 7
**Confidence:** 5

**Review:**

Transformer based bidirectional LMs pre-trained using Masked Language Model loss typically share input and output token embeddings. This paper makes an interesting investigation about decoupling input and output embeddings and gains which can be obtained out of this decoupling. In particular, this paper shows that the pre-training performance of transformers and the transferability of the learned representations can be improved by increasing the dimension of output embeddings while reducing the dimension of input embeddings. Better performance while pre-training and improved transferability further helps the performance while finetuning on downstream tasks. Parameters saved while reducing the dimensions of input embeddings can be re-invested into increasing the depth or width of the transformer layers. I believe that the findings in this paper are going to be useful in practice to the general NLP community working on Transformers and Multilingual problems.

Weak Points:
1. In table 3, (E_in=768,E_out=128) is just 0.1 point worse on average than (E_in=128, E_out=768).  Yet, authors draw some conclusions on Page 4 based on this table, without reporting any variance or statistical significance tests.
    E.g.
    >  the model pretrained with a larger output embedding size slightly outperforms the comparison method on average despite
    having 77M fewer parameters during fine-tuning

    > Reducing the input embedding dimension saves a significant number of parameters at a noticeably smaller cost to accuracy
    than reducing the output embedding size.

I would request the authors to report the variance in Table 3 and also in Table 9. Otherwise, it’s hard to rely on conclusions drawn from such minor differences in performance.


Strong Points:
Strong empirical results.
1. Table 4 clearly shows that increasing the dimension of output embeddings improves transfer to downstream tasks while having the same number of trainable parameters during the finetuning stage.
2. Table 6 shows that reinvesting the parameters saved from reducing the dimension of input embeddings into additional transformer layers yield improved performance. Further, RemBERT performs at par with XLM-R while having a comparable number of trainable parameters during the finetuning stage.
3. Sufficient amount of analysis in Section 6 to establish the usefulness of having higher dimensional output embeddings for improved pre-training and better transferability of the learned representations across tasks.

Other comments/questions:
1. Shouldn’t #PT params in table 2 for the “Decoupled” method be more than #PT params for the “Coupled" method ?
2. More figures like Figure 1, on other tasks in addition to XNLI, would be really helpful in making the observations more conclusive.

---

> ### Author Response · Authors · 2020-11-18
> **We adjusted our claim regarding Table 3.**
>
> > “Yet, authors draw some conclusions on Page 4 based on this table, without reporting any variance or statistical significance tests.”
>
> The average variance for the scores of the $E_{\textrm{in}} = 768, \ E_{\textrm{out}} = 128$ and the $E_{\textrm{in}} = 128, \ E_{\textrm{out}} = 768$ model in Table 3 is 0.8 and 0.05 respectively across three runs. Note that our main point in this paragraph does not require the improvement of the smaller model to be statistically significant. Instead, our main finding here is the fact that the smaller model is **not significantly worse** than the larger one. This is surprising in our opinion, given that the smaller model has 77M (or 43%) fewer parameters during fine-tuning. We will clarify this in the revised version, highlighting that not the magnitude of the improvement of the smaller model is important but that it is competitive with the larger model with a much smaller number of parameters.
>
> For the probing experiments in Table 9, we found that the standard deviation between experiments to be very small, agreeing with the findings of Tenney et al. (2019) who reported about 0.2 (or variance of 0.04) for most tasks.
>
> > "Shouldn’t #PT params in table 2 for the “Decoupled” method be more than #PT params for the “Coupled" method ?"
>
> We have fixed the typo that misspecified the number of pre-training parameters in Table 2. Thank you for catching the typo.
>
> > "More figures like Figure 1, on other tasks in addition to XNLI, would be really helpful in making the observations more conclusive."
>
> Per your suggestion, we will run additional experiments to generate more figures like Figure 1. We will update the manuscript with the additional figures.
>
> If the reviewer feels like we adequately answered their concern, given that this is the only negative point noted by them, we’d like to encourage them to potentially reconsider their score. Otherwise we’d be happy to continue the conversation.

---

### Author Response · Authors · 2020-11-20
**We updated the paper and answered each reviewer individually.**

We thank the reviewers for their time and thoughtful feedback. All reviewers noted that our empirical results were strong and well supported by analysis. Reviewers further highlighted that the paper is well written and strongly motivated (R1, R3), that our strategy is novel (R3) and simple yet effective (R4), and that the findings will be “useful in practice to the general NLP community” (R2) and “promote future model design and the understanding of the transformer” (R3).

We respond to individual reviews in a reply to the corresponding review. We have uploaded a new version fixing typos and adding information reviewers asked for (see the individual replies for more information).

---

### Decision · Program_Chairs · 2021-01-07
**Final Decision**

**Decision:**

Accept (Poster)

**Comment:**

This paper makes a thorough investigation on the idea of decoupling the input and output word embeddings for pre-trained language models.  The research shows that the decoupling can improve the performance of pre-trained LMs by reallocating the input word embedding parameters to the Transformer layers, while further improvements can be obtained by increasing the output embedding size.  Experiments were conducted on the XTREME benchmark over a strong mBERT.  R1&R2&R3 gave rather positive comments while R4 raised concerns on the model size.  The authors gave detailed response on these concerns but R4 still thought the paper is overclaimed because the experiments were only conducted in a multilingual scenario.